# Phosphorylation of Human Polyomavirus Large and Small T Antigens: An Ignored Research Field

**DOI:** 10.3390/v15112235

**Published:** 2023-11-09

**Authors:** Ugo Moens, Sara Passerini, Mar Falquet, Baldur Sveinbjørnsson, Valeria Pietropaolo

**Affiliations:** 1Department of Medical Biology, Faculty of Health Sciences, University of Tromsø—The Arctic University of Norway, 9037 Tromsø, Norway; marfalquet@gmail.com (M.F.); baldur.sveinbjornsson@uit.no (B.S.); 2Department of Public Health and Infectious Diseases, “Sapienza” University of Rome, 00185 Rome, Italy; sara.passerini@uniroma1.it

**Keywords:** human polyomavirus, large T antigen, Merkel cell polyomavirus, protein kinase, small t antigen, SV40

## Abstract

Protein phosphorylation and dephosphorylation are the most common post-translational modifications mediated by protein kinases and protein phosphatases, respectively. These reversible processes can modulate the function of the target protein, such as its activity, subcellular localization, stability, and interaction with other proteins. Phosphorylation of viral proteins plays an important role in the life cycle of a virus. In this review, we highlight biological implications of the phosphorylation of the monkey polyomavirus SV40 large T and small t antigens, summarize our current knowledge of the phosphorylation of these proteins of human polyomaviruses, and conclude with gaps in the knowledge and a proposal for future research directions.

## 1. Introduction

Posttranslational modification is a pivotal mechanism regulating the function of a protein. One such posttranslational modification is the reversible phosphorylation/dephosphorylation of target proteins controlled by the opposite action of protein kinases and protein phosphatases, respectively [1,2]. The human genome encodes 518 protein kinases, while the human protein phosphatome is composed of 189 known and predicted protein phosphatase genes [3,4]. Phosphorylation occurs at serine (Ser, S), threonine (Thr, T), and tyrosine (Tyr, Y) residues, and it is estimated that about 30% of cellular proteins are phosphorylated at least at one residue [5]. Of these three residues, Ser is the most often phosphorylated, followed by Thr and Tyr [6]. The phosphorylation status of a protein provides a way to control its activity, stability, subcellular localization, and interaction with itself or other proteins [7,8]. Therefore, protein phosphorylation plays a crucial role in cellular processes such as signal transduction, transcription regulation, cell cycle progression, metabolic regulation, autophagy, and apoptosis [9,10,11,12]. Aberrant expression or activity of many protein kinases and protein phosphatases are associated with malignant and non-malignant diseases, underscoring the pivotal role of these enzymes [13,14].

Historically, phosphorylated residues were identified by using in vitro or in vivo ^32^P-labeled proteins. ^32^P-labeled peptides were separated through polyacrylamide gel electrophoresis or high-performance liquid chromatography and detected using autoradiography or scintillation counting. Edman degradation was then used to sequence the radiolabeled peptides to determine the phosphorylated residue(s). More recently, mass spectrometry is the method of choice for characterizing phosphorylated proteins. The protein of interest is enzymatically digested into peptides and analyzed using a mass spectrometer, in which the instrument records the mass-to-charge ratios of the various peptides. To identify a phosphorylated peptide, its mass-to-charge values are compared to the expected mass-to-charge values of peptides. Phosphorylated peptides have an increased mass of n × 79.9663 Da (i.e., mass of phosphate group; n = number of phosphates in the peptide). For reviews, see [15,16].

Among substrates for protein kinases and protein phosphatases are proteins from both RNA and DNA viruses, and their phosphorylation forms an essential step in the viral life cycle [17,18,19,20,21,22]. Phosphorylation/dephosphorylation is mediated by cellular protein kinases/phosphatases, but some viruses encode their own protein kinases and protein phosphatases [23,24,25,26,27,28]. Proteins of the DNA virus family *Polyomaviridae* are a target for protein kinases. Polyomaviruses are non-enveloped viruses with an icosahedral capsid surrounding a circular double-stranded genome of approximately 5000 base pairs. Polyomavirus infections have been described in many vertebrates, and polyomavirus DNA has also been reported in invertebrates [29,30]. According to the International Committee on Taxonomy of Viruses, this family is divided into eight genera: Alphapolyomavirus, Betapolyomavirus, Gammapolyomavirus, Deltapolyomavirus, Epsilonpolyomavirus, Zetapolyomavirus, Etapolyomavirus, and Thetapolyomavirus [ICTV; email ratification March 2023 (MSL #38)]. Sixteen different PyVs have been described in humans: SV40 (or Betapolyomavirus macacae), BKPyV (Betapolyomavirus hominis), JCPyV (Betapolyomavirus secuhominis), Karolinska Institute polyomavirus (KIPyV or Betapolyomavirus tertihominis), Washington University polyomavirus (WUPyV or Betapolyomavirus quartihominis), Merkel cell polyomavirus (MCyPyV or Alphapolyomavirus quintihominis), HPyV6 (Deltapolyomavirus sextihominis), HPyV7 (Deltapolyomavirus septihominis), Trichodysplasia spinulosa-associated polyomavirus (TSPyV or Alphapolyomavirus octihominis), HPyV9 (or Alphapolyomavirus nonihominis), HPyV10/Malawi polyomavirus (MWPyV or Deltapolyomavirus decihominis), STLPyV (Saint Louis polyomavirus or undecihominis), HPyV12, New Jersey polyomavirus (NJPyV or Alphapolyomavirus terdecihominis), Lyon IARC polyomavirus (LiPyV), and QPyV (unclassified Deltapolyomavirus) [31,32,33,34,35,36,37,38,39,40,41,42,43,44]. However, not all are genuine human polyomaviruses (HPyVs). SV40 has Rhesus macaques as its natural host [45,46,47,48], whereas HPyV12 seems to infect shrews [49] and LiPyV is more likely a feline PyV because seropositivity in cats is 92.5%, compared to only 2.3% in humans [50]. QPyV DNA has only been detected in a few human specimens, and serological studies are lacking. Because of its high sequence identity with HPyV6 and HPyV7 (67% and 80%, respectively), QPyV is classified as a Deltapolyomavirus [44,51]. In this review, HPyV12 and LIPyV will not be considered as true HPyVs, whilst QPyV is included, although only two reports have described the presence of viral DNA in human samples [44,51].

The HPyV genome encodes regulatory and structural proteins. The structural proteins are the capsid proteins VP1, VP2, and in most cases VP3, whereas the regulatory proteins include LTAg and sTag [52,53]. Some HPyVs encode additional regulatory proteins such as JCPyV T’_135_, T’_136_, and T’_165_ [54] and ALTO protein for MCPyV and TSPyV [55,56]. In contrast to LTAg and sTAg, the functions of these additional regulatory proteins are poorly understood. Capsid proteins of several members of the *Polyomaviridae* family have been shown to be phosphorylated, which is essential for viral attachment to the host cell and propagation, for example [57,58]. The biological implications of LTAg have been mostly studied for SV40 and will be summarized in this review. Potential and proven phosphorylation sites in human polyomavirus LTAg and sTAg and their impact on their functions will also be discussed.

## 2. Phosphorylation of Large T Antigen

### 2.1. Functions of SV40 LTAg

SV40 LTAg consists of 708 amino acids and is a multifunctional protein. The structure and different functions have been extensively and excellently reviewed before and will therefore only be briefly mentioned in this review [59,60,61,62,63,64,65,66]. Because of the sequence homology with LTAg from HPyVs, it is assumed that HPyV LTAg exhibits similar functions. The LTAg of SV40 is absolutely required for replication of viral DNA. LTAg is the only viral protein involved in replication of the viral genome. LTAg binds to repeated 5′-GAGGC-3′ motifs in the non-coding control region, which consists of the origin of replication (ORI) and the transcriptional control region. The core ORI consists of LTAg binding site I with tandem repeats of the GAGGC motif, the early palindrome, which also contains two GAGGC motifs in the opposite direction, a central palindrome encompassing binding site II with double tandem GAGGC repeats in opposite direction, and an AT-rich region [66,67,68]. DNA replication is initiated when two hexamers of LTAg attach in a head-to-head orientation at site II in the ORI. LTAg recruits cellular enzymes involved in DNA replication, and the intrinsic helicase activity of LTAg helps unwind double-stranded DNA. Following initiation, the double hexamers separate from each other and move in opposite directions to replicate viral DNA in a bidirectional manner [59,66,68]. As will be outlined in Section 2.3.2 phosphorylation of LTAg regulates the role of LTAg in viral DNA replication. In addition, LTAg is involved in regulating transcription of the early and late genes. LTAg can also transform cells in vitro and induce tumors in animal models [59,63,65,66]. SV40 DNA, RNA and the LTAg protein have been detected in human cancers, such as mesothelioma and brain cancer tissue, but SV40’s role in human malignancy remains to be discussed [69,70,71].

### 2.2. Putative and Proven Phosphorylation Sites in SV40 LTAg

The phosphorylation prediction algorithm program Netphos 3.1 was used to identify putative phosphoacceptor sites and protein kinases that mediate their phosphorylation [72,73]. The results are summarized in Appendix A and predicted more than 50 phosphoacceptor sites spread throughout the LTAg sequence. However, studies have shown that SV40 LTAg is only phosphorylated in the N- and C-terminal region of the protein at residues S106, S111, S112, S120, S123, T124, S639, S665, S667, S676, S677, S679 and T701, whilst the central part appears to be unphosphorylated (see Table 1; [74,75,76]), although not all studies confirmed phosphorylation of S676 [74]. Phosphorylation could be mediated in vitro and/or in vivo by casein kinases 1 and 2 (CKI and CKII), DNA-dependent protein kinase (DNAPK), glycogen synthase kinase-3 (GSK3), cyclin-dependent kinase 1 (CDK1 or previously known as cell division cycle *protein* 2 homolog cdc2), and ataxia-telangiectasia-mutated (ATM). The protein kinases cAMP-dependent protein kinase (or protein kinase A, PKA), cGMP-dependent protein kinase (or PKG), PKC, protein kinase N1 (PKN1 or protein kinase C-related kinase 1, PRK1), and PKN2 failed to mediate in vitro phosphorylation of SV40 LTAg [75,77]. The biological effects of mutation in the phosphoacceptor sites of SV40 are summarized in Table 1 and will be discussed in detail in the following section.

### 2.3. Biological Implications of SV40 LTAg Phosphorylation

#### 2.3.1. SV40 LTAg Phosphorylation and Subcellular Distribution

SV40 LTAg, like all eukaryotic proteins, is synthesized in the cytoplasm, but it is mainly relocalized to the nucleus. A minor fraction associates with the plasma membrane and is exposed to the cell surface [88]. Nuclear import is regulated by a nuclear localization report (NLS) with the sequence PKKKRKV (residues 126–131) and the nuclear transport factor importin [89]. Early studies showed differences in the phosphorylation of cytoplasmic and nuclear LTAg, suggesting that in addition to the NLS, phosphorylation may affect the subcellular localization of LTAg [90]. Indeed, it was found that phosphorylation either inhibited or stimulated nuclear import depending on the residues phosphorylated. Phosphorylation of S106 by CKI or GSK3 inhibited nuclear import by approximately 50% compared to non-phosphorylated LTAg [75]. The group of Jans showed that the binding of the retinoblastoma family member p110 (RB1) to SV40 LTAg inhibited the nuclear import of LTAg, and phosphorylation of S106 within LTAg enhanced the interaction with RB1 and further reduced the nuclear import. However, RB1 binding did not impair the NLS of LTAg interacting with importin [91]. It should be mentioned that these studies were performed with GFP fusion proteins containing aa 87 (respectively 102 and 110) −135 of LTAg, so that the effect of phosphorylation in the C-terminal part on RB1-mediated cytoplasmic retention was not examined. Phosphorylation of T124 by CDK1 also had a negative effect on the nuclear import [81,92,93,94]. Nuclear accumulation of the LTAg T124A mutant increased compared to wild-type LTAg, and the T124D mutant reduced the nuclear import of LTAg [95], as reviewed in [17]. Fulcher et al. showed that the BRCA1 binding protein 2 (BRAP2) binds LTAg and blocks nuclear import. Phosphorylation of T124 and an intact NLS are necessary for the interaction between LTAg and BRAP2, and this association prevents LTAg from entering the nucleus. The T124A mutation abrogates binding of BRAP2 to LTAg, whereas T124D increases the affinity of BRAP2 for LTAg, explaining the increased or reduced nuclear import of LTAg, respectively [95]. Phosphorylation of S111/S112 by CKII and S120 by DNAPK enhanced the nuclear import, and mutation/deletion of S111/S112 to prevent phosphorylation decreased the nuclear import rate [81,92,93,94,96]. The LTAg S112D mutant displayed enhanced nuclear import compared to wild-type LTAg [92,97], as reviewed in [17]. CKII-mediated phosphorylation of S111/S112 increased the affinity of importin for LTAg approximately 50-fold, whilst phosphorylation of S120 by DNAPK enhanced importin binding by 40% and 100-fold in synergy with CKII-catalyzed phosphorylation of S111/S112 [93], as reviewed in [17]. S112 seems to be the main site of phosphorylation by CKII [98]. The role of S120 in nuclear import was elucidated by generating the LTAg mutants S120A and S120D. Both mutations reduced nuclear accumulation by 70% and 40%, respectively, compared to wild-type LTAg, but the S120D mutant showed a faster import rate than wild-type LTAg [94]. The participation of S123 in nuclear localization was also examined but revealed that mutation of S123 did not influence the nuclear transport rate of SV40 LTAg [81]. Taken together, the phosphorylation of LTAg allows finetuning of its subcellular localization and may therefore play an import role in the viral life cycle because this protein is required for viral DNA replication and transcription, which occur in the nucleus [61,65,66].

#### 2.3.2. SV40 LTAg Phosphorylation and Replication

A major role of LTAg involves viral DNA replication and requires that LTAg binds to the ORI. Upon binding to the GAGGC motifs in site II of the ORI, LTAg monomers assemble into hexamers and subsequently into double hexamers [99,100]. It was shown that the phosphorylation pattern of unbound and DNA-associated LTAg is different [90]. This suggested that phosphorylation may affect the binding of LTAg to DNA. Indeed, treatment of LTAg with potato acid phosphatase, which removes Ser- and Thr-bound phosphate groups, reduced LTAg binding to SV40 DNA [101], whereas treatment with alkaline phosphatase, which removes Ser-bound phosphate groups, increased DNA binding [102,103]. Simmons and colleagues found that alkaline phosphatase treatment of immunoprecipitated LTAg increased its in vitro DNA binding activity, whereas phosphorylation of the N-terminal region (residues 106–124) reduced in vitro DNA binding [104]. Other groups reported that treatment with calf intestinal alkaline phosphatase, which removes Ser-bound phosphates, enhanced LTAg’s ability to stimulate DNA replication but had no effect on DNA binding nor on the ATPase activity of LTAg [105,106,107]. McVey and co-workers demonstrated, using bacterial-expressed LTAg, that T124 was a phosphoacceptor site for CDK1. This in vitro T124 phosphorylated purified LTAg, bound better to SV40 ORI, and stimulated SV40 DNA replication compared to unphosphorylated purified LTAg under in vitro conditions [86]. In vivo phosphorylation of LTAg at T124 by CDK1 also stimulated viral DNA replication. Phosphorylation of T124 by CDK1, a master regulator of the cell cycle [108], stimulated LTAg binding to the ORI and DNA unwinding, but had no effect on DNA polymerase α binding or the and helicase and transcriptional activity of LTAg [109]. Later studies showed that phosphorylation of T124 had no effect on the formation of the first hexamer but promoted double hexamer formation [110]. This was confirmed by the fact that the LTAg T124A mutant, although able to bind the SV40 ORI, was defective in the initial opening of the duplex at the ORI, possibly because of distorted double hexamer formation [111]. The ability of LTAg phosphorylated at T124 to enhance SV40 DNA replication seems contradictory to the finding that phosphorylation of this residue inhibits the nuclear import of LTAg, as described in Section 2.3.2. Phosphorylation of T124 may occur in a cell-cycle-dependent manner because CDK1, which mediates phosphorylation of this site, is regulated in a cell-cycle-dependent manner [108]. The observation that phosphorylation of LTAg in the cytoplasm was relatively stable, whereas phosphorylation of LTAg in the nucleus exhibited a higher turnover rate, may restrict the time during which LTAg contributes to viral DNA replication and regulate the switch to the late phase in the viral life cycle, which includes expression of the capsid proteins and their assembly with viral DNA into new virus particles [112].

Other phosphorylation events are involved in viral DNA replication by LTAg. Vishrup et al. demonstrated that PP2A removed phosphates from phosphoSer 120, 123, 677, and perhaps 679, but not phosphoThr residues in purified ^32^P-labeled LTAg, and purified PP2A preferentially stimulated SV40 DNA replication in extracts from early G_1_ phase cells [83,113]. Dephosphorylation of LTAg by PP2A promoted binding of the second LTAg hexamer to the SV40 ORI, thereby stimulating DNA unwinding [114]. CKI phosphorylated residues S120, S123, S676, and S679 (Table 1). CKI-catalyzed phosphorylation of S120 and S123 inhibited viral DNA replication, whereas phosphorylation of S676 and S679 were not required for the inhibition of replication [77]. Dephosphorylation of phosphoS120 and phosphoS123 by PP2A were required for full activation of LTAg replication potential in vitro [77,83,86,105]. Phosphorylation occurred on full-length LTAg (FL-LTAg) but not on an N-terminal fragment containing the first 259 amino acids. The authors suggested that CKI-mediated phosphorylation required a three-dimensional structure positioning the N-terminal and C-terminal domains in proximity [77]. This assumption is underscored by the finding that mutations in the phosphoacceptor sites in the N-terminal part reduced the phosphorylation of residues in the C-terminal domain and vice versa. For example, mutation of S106 prevented the phosphorylation of S639, and mutation of S677 reduced the phosphorylation of S120 and S123 [74]. In agreement with this mechanism, Scheidtman et al. proposed that phosphorylation of Ser-677 is required for the subsequent phosphorylation of S120 and S123 [74]. S120 is also the substrate for ataxia-telangiectasia-mutated (ATM). However, in contrast to CKI-mediated phosphorylation, phosphorylation of S120 by ATM activated SV40 replication in CV1 African green monkey cells [85]. Moreover, an SV40 mutant encoding LTAg S120A displayed reduced replication in monkey cells compared to wild-type SV40 [78,85]. The opposite effect of S120 phosphorylation on SV40 replication may be explained by the experimental conditions (in vitro versus in vivo). Alternatively, the phosphorylation of S120 by CKI and ATM may be timeously regulated and result in the stimulation or inhibition of viral DNA replication in a time-dependent manner. Hence, the timing of S120 phosphorylation may be crucial to prevent premature SV40 DNA replication [85]. Interestingly, SV40 sTAg prevented dephosphorylation of LTAg by PP2A, preferentially of S120 and S123 [115]. Hence, sTAg may regulate the replication activities of LTAg.

#### 2.3.3. SV40 LTAg Phosphorylation and Protein Interaction

SV40 LTAg can interact with several proteins (see Appendix A in [53]). One of the interaction partners is F-Box and WD Repeat Domain Containing 7, E3 Ubiquitin Protein Ligase (Fbxw7), which is part of the phosphorylation-dependent ubiquitination process in proteasomal degradation of substrates [116]. Fbxw7 was shown to bind to SV40 LTAg in a phosphoT701-dependent manner [117]. As mentioned above, phosphorylation of SV40 LTAg at T124 is required for the interaction with BRAP2, and this association inhibits nuclear import of LTAg [95]. Whether binding of SV40 LTAg to other proteins depends on phosphorylation was not examined.

#### 2.3.4. SV40 LTAg Phosphorylation and Transformation

Phosphorylation of specific residues in the C-terminal part had an effect on the transforming activities of LTAg (Table 1). The mutations S639A and S67A enhanced the transforming activity in Rat-1 cells, whereas the substitutions S667A and S679A had the opposite effect [74,78]. In the same cell line, LTAg S106F and S189D displayed reduced transforming activity [74,118]. The latter site is not a known phosphorylation site, although it is part of a weak PKA and RSK consensus motif (Appendix A), suggesting that phosphorylation is not the only mechanism to control the transforming activity of LTAg. Another study, performed on Rat-2 cells, reported that the LTAg S106A mutant displayed comparable transforming activity to wild-type LTAg (Table 1; [78]).

In conclusion, the phosphorylation state of SV40 LTAg has an impact on its subcellular localization and on viral DNA replication. The phosphorylation profile of residues S106, S111/S112, and T124 controls the nuclear import of LTAg, whereas phosphorylation of T124 is crucial for viral DNA replication. LTAg can bind to p53 and pRb, and this will induce cell progression, resulting in activation of CDK1 and PP2A. CDK1 mediates the phosphorylation of T124, whereas PP2A dephosphorylates Ser residues. The role of phosphorylation of the C-terminal Ser residues could be to repress early transcription and stimulate late transcription, but this remains to be determined (and is reviewed in [119]). Despite its role in Fbxw7 interaction (Section 2.3.3), the biological implication of T701 phosphorylation remains elusive because the LTAg T701A showed no differences to wild-type LTAg concerning virus viability, replication, transforming ability, nuclear localization, site I and site II binding, expression levels, and ATPase activity (Table 1; [74,76,78,79]).

### 2.4. Phosphorylation of HPyV LTAg

#### 2.4.1. Phosphorylation of Human Betapolyomavirus LTAg

Very few studies have investigated the phosphorylation of LTAg of HPyVs. Tryptic phosphopeptide mapping revealed that JCPyV LTAg is phosphorylated at Ser and Thr but not Tyr residues [120]. Similar to SV40 LTAg, phosphorylation regions were mapped in the N-terminal and C-terminal domains, with the majority of phosphorylation in the N-terminal part. The exact residues were not determined, but by using the LTAg T664A mutant, Swenson and Frisque found that T664 was not phosphorylated. The authors also found that the phosphorylation pattern is slightly different in JCPyV LTAg isolated from human and rat cells. The protein kinases that mediated JCPyV LTAg phosphorylation have not been identified, but CKI can phosphorylate LTAg in vitro [120]. Residues S248 and S640 are predicted as weak CKI phosphoacceptor sites by the Netphos 3.1 algorithm, but CKI-mediated phosphorylation of these amino acids has not been experimentally proven. Based upon sequence homology with SV40 LTAg, one or several of the JCPyV residues S114, S121, S124, and T125 were predicted to be phosphoacceptor sites [121]. Later studies examined the possible phosphorylation and biological role of T125 in JCPyV LTAg, which corresponds to T124 in SV40 LTAg, by generating non-phosphorylable and phosphomimicking mutants [122]. The T125A mutant protein was less stable than wt LTAg, whereas the stability of the T125A variants of T’_135_, T’_136_, and T’_165_ was comparable to wild-type T’ variants. The stability of the T125D LTAg proteins was similar to the wild-type proteins. T125A substitution slightly reduced the ability of mutant LTAg proteins to bind to the retinoblastoma proteins p107 (RBL1) and p110 (RBL2) and failed to release transcription factor E2F from the E2F:RB protein complex. T125A mutants, on the other hand, bound to RBL1 and RBL2 more efficiently compared to wild-type LTAg and released E2F. T125A LTAg mutants had no transforming activity, whereas the T125D protein could transform Rat-2 cells. JCPyV expressing LTAg T125A or T125D failed to replicate in primary human fetal glial cells [121,122]. These results indicate that phosphorylation of T125 has an effect on the biological properties of JCPyV LTAg proteins. Although prediction algorithms can be useful tools to predict phosphosites, of the putative sites S114, S121, S124, and T125 [121], only S121 was suggested as a possible phosphoacceptor site by Netphos 3.1 (Appendix A). T664 is a predicted weak phosphoacceptor site for ATM and CKII (Appendix A). However, substitution of T664 into nonphosphorylable alanine had no effect on JCPyV replication in primary human fetal glial cells, and this residue was shown not to be phosphorylated [121].

Only one study reports the functional implication of JCPyV LTAg phosphorylation. Beta-transducin-repeat-containing proteins (BTRCPs or β-TrCPs) are components of the Skp1-Cul-F-box (SCF) protein E3 ubiquitin ligase complex, which plays a role in proteasomal degradation. They interact with the consensus motif DpSGX_2–4_pS on their substrates [123]. BTRCP1 and BTRCP2 (or FBXW1B) were shown to bind to LTAg of JCPyV in a phosphorylation-dependent manner and involved residues 639–645 (DSGHGSS) [124]. BTRCP1-LTAg interaction required phosphorylated S640 and to a lesser extent phosphorylated S644 of JCPyV LTAg, whereas only phosphorylated S640 seemed to be necessary for interaction with BTRCP2. The JCPyV LTAg-BTRCP interaction may therefore affect the proteasomal pathway and contribute to the pathogenicity of JCPyV. The sequence DSGHGSS is completely conserved in BKPyV LTAg (amino acids 640–646), but not in SV40 (DSGHETG; residues 656–662). SV40 LTAg was unable to bind BTRCPs, but the interaction with BKPyV LTAg was not investigated [124]. The BTRCP binding consensus motif DSGX_2–4_S is present in the LTAg of the non-Betapolyomaviruses MCPyV (DSGTFSQ; residues 811–817) and HPyV6 (DSTQESG; residues 656–662), but not in the LTAg of other HPyVs. Dysfunction of the proteasomal pathway is associated with cancer, and a role for BKPyV, MCPyV, and HPyV6 in malignancy is known or emerging [125,126,127], suggesting that disturbance of the proteasomal pathway by LTAg of these HPyVs may be a contributing factor in HPyV-induced oncogenesis.

BKPyV is another well-studied Betapolyomavirus. Although phosphorylation of its capsid proteins has been examined and phosphorylation sites have been identified [58,128], little is known about the phosphorylation of its LTAg. Immunoprecipitation of ^32^P-labelled cells demonstrated that BKPyV is a phosphoprotein, but the phosphoacceptor sites and protein kinases that mediate phosphorylation remain unknown [129].

The phosphorylation of LTAg of the other members of Betapolyomavirus, KIPyV and WUPyV, has not been studied. Of the proven phosphorylation sites in SV40 LTAg (i.e., S106, S111, S112, S120, S123, T124, S639, S665, S667, S676, S677, S679, and T701), S112 (S114 in BKPyV and JCPyV, S117 in KIPyV, and WUPyV) and T124 (T126, T125, T133, and T139 in BKPyV, JCPyV, KIPyV, and WUPyV, respectively) are conserved among LTAg of Betapolyomavirus (Figure 1A and Appendix A). The SV40 S112 corresponding residue in these HPyVs is a putative target for CKII (BKPyV and JCPyV), CDK5 and GSK3 (KIPyV), and CKI, GSK3, and p38^MAPK^ (WUPyV). T124 is part of a conserved TPPKKK motif and is a predicted phosphoacceptor for p38^MAPK^, CDK5, GSK3, and PKC (Appendix A). SV40 S123 is only conserved in LTAg of BKPyV (S125) and JCPyV (S124). The C-terminal S639 and T701 residues are also only present in BKPyV (S641 and T691) and JCPyV (S640 and T684) LTAg. Further studies are required to confirm whether these are genuine phosphoacceptor sites and whether their phosphorylation affects the functions of LTAg.

#### 2.4.2. Phosphorylation of Human Alphapolyomavirus LTAg

##### Phosphorylation of Merkel Cell Polyomavirus Full-Length LTAg

Merkel cell polyomavirus (MCPyV), a causative agent of Merkel cell carcinoma (MCC), was first isolated in 2008 [35]. The virus is present in about 80% of all MCC samples examined and is always integrated in the host genome. MCPyV encodes an 817-amino-acid-long LTAg (hereafter referred to as full-length LTAg or FL-LTAg). Another hallmark of MCPyV-positive MCCs is the presence of a nonsense mutation in the LTAg gene resulting in expression of a C-terminal truncated LTAg [130]. The size of the truncated LTAg (hereafter referred to as tLTAg) described so far ranges from 163 residues [131] to 538 amino acids [unpublished; accession number KJ022619]. Because phosphorylation of both FL- and tLTAg has been studied and sometimes opposite effects have been reported, they are discussed in first two separate subsections in Section 2.4.2, respectively. FL-LTAg has a statistical overrepresentation of Ser residues, especially in the N-terminal half of the protein. Thr is also slightly overrepresented in this region, suggesting that many of these sites may function as phosphoacceptor sites (Appendix A). Many of these sites are putative phosphorylation sites for protein kinases (Appendix A). Because of its pathogenic property, the phosphorylation of MCPyV LTAg has been more intensively studied than any other HPyV. A restricted number of putative phosphorylation sites have been experimentally confirmed. Mass spectrometry analysis of FL-LTAg showed that at least 17 Ser and Thr residues were phosphorylated in the amino acids 1–278 [132]. These include S100, S134, S147, T192, S239, S254, T257, S265, and T271. In addition, at least one of the residues 172–179 (TSSSGSSS), S202, S203 and T205, 217–220 (SLSS), S225, S226, and S227 is phosphorylated [132]. Another study confirmed the phosphorylation of T271 and additionally identified the phosphorylation of T295 and T299 in FL-LTAg expressed in the human embryonal kidney cell line HEK293 [133].

So far, only two protein kinases that can phosphorylate MCPyV LTAg have been identified. You and collaborators showed that ATM phosphorylates S816A, and this partially reverses inhibition of the growth of the human papillomavirus-negative cervical cancer cell line C33A compared to wild-type FL-LTAg and reduces apoptosis [134]. As virus-positive MCCs express C-terminal truncated LTAg lacking this residue, the biological importance of this phosphorylation event does not seem to be relevant in MPyV-positive MCCs. We have previously shown that S203 and S265 are in vitro phosphorylated by cAMP-dependent protein kinase (or PKA) [135]. Mutations in these residues repressed the transcriptional activity of FL-LTAg, and single Ser into Ala and Ser into Asp mutations had no effect on the transcriptional activity [135].

Mutation studies have shown that MCPyV LTAg with either S96A, S134A, S179A, S186A, T192, T268A, T271A, T299A, T309A, S597A, S715A, or S816A had no effect on the half-life of the full-length protein [136]. Mutation S142A increased the half-life of FL-LTAg and reduced the interaction with β-TrCP [137]. S147A also stabilized FL-LTAg, which did not affect the transcriptional activity of the protein, but this mutant could not stimulate viral replication [136,137]. The S220A mutation increased the half-life, viral DNA replication, and transcriptional activity, but impaired RB1 binding and reduced the interaction with SCF protein E3 ubiquitin ligase component S-phase kinase associated protein 2 (SKP2) [132,136,137,138]. FL-LTAg was shown to bind to the hVam6p (or VPS39) subunit of the HOPS complex, a protein that promotes clustering and fusion of late endosomes and lysosomes [139], through residues 204–218. Mutations T205A, Y206A, and T208A in putative phosphorylation sites did not interfere with this interaction, suggesting that phosphorylation of MCPyV LTAg is not required for the association with hVam6p [140]. The FL-LTAg mutants T271A, T297A, and T299 were tested for their effect on ORI binding and replication of viral DNA. The T291A substitution had no effect, whereas T297A increased and T299A reduced LTAg binding to the ORI and impaired replication [133].

##### Phosphorylation of Merkel Cell Polyomavirus Truncated LTAg

Our group demonstrated that tLTAg stimulated the MCPyV promoters, and Ser into Ala mutations in the PKA phosphorylation sites S203 and S265 inhibited tLTAg’s transcriptional activity by 50–80% compared to non-mutated tLTAg. Single Ser into Asp mutations restored the transcriptional activity of tLTAg [135]. Studies by another group showed that single non-phosphorylable or phosphomimicking mutations in tLTAg of T192, S202, S203, T205, S217, or S239 had no effect on the growth of the MCC cell line MKL1 [132]. S219A or S220A, but not S219E or S220E, tLTAg partially inhibited (20% and 60%, respectively) the growth of these cells [132]. S239A tLTAg had increased half-life compared to unmutated tLTAg and reduced interaction with Fbxw7 [136,137]. As discussed in Section 2.3.1, the phosphorylation of residues adjacent to the NLS played a role in the nuclear import of SV40 LTAg. The MCPyV LTAg possesses the NLS motif RKRK (residues 277–280). The S246A/S247A/S254A/T25 7A/S265A/T271A mutant had no effect on the nuclear translocation of tLTAg (aa1–334) in the mouse fibroblast NIH3T3 cell line [132]. The effect of these substitutions in context of FL-LTAg was not investigated.

##### Phosphorylation of Human Alphapolyomavirus LTAg

To the best of our knowledge, the phosphorylation of LTAg of the other Alphapolyomavirus species TSPyV, HPyV9, and NJPyV has not been examined. SV40 T124, which was also conserved in the Betapolyomavirus, is also conserved in the Alphapolyomavirus (T199 in MCPyV, T174 in TSPyV, T170 in HPyV9, and T205 in NJPyV; Figure 1A). This may suggest that the T124 corresponding residue in LTAg of HPyV is a genuine phosphoacceptor site.

#### 2.4.3. Phosphorylation of Human Deltapolyomavirus LTAg

Studies investigating the phosphorylation status of Deltapolyomavirus LTAg are lacking. Two proven SV40 phosphoacceptor sites are conserved in LTAg of Deltapolyomavirus: S112 and T124 (Figure 1A and Appendix A). The corresponding S112 residue is S118 in HPyV6, HPyV7, and QPyV and S114 in HPyV10 and STLPyV, whilst T124 is T139 in HPyV6, T144 in HPyV7, T140 in QPyV, T149 in HPyV10, and T136 in STLPyV.

In conclusion, knowledge about the phosphorylation of HPyV LTAg is almost non-existent. The SV40 T124 residue is conserved in LTAg of all HPyVs, whereas SV40 S112 is conserved in Beta- and Deltapolyomaviruses. T124 is part of a conserved TPP (K/R)_4_ motif found in the LTAg of all HPyVs. However, neither their phosphorylation and the protein kinases that catalyze this posttranslational modification have been demonstrated, nor has the biological function of their possible phosphorylation been revealed.

## 3. Phosphorylation of sTAg

### 3.1. Function of SV40 sTAg

SV40 sTAg is generated through alternative splicing of the early transcript, and the N-terminal 82 amino acids are identical to LTAg [61]. sTAg can transactivate different promoters and exerts an auxiliary role in the transformation of certain cell types by LTAg, but in some cells, sole expression of sTAg is sufficient for transformation [141,142]. The major contribution of sTAg in transformation is its ability to inhibit protein phosphatase PP2A activity, thereby perturbing the phosphorylation pattern of cellular proteins, but also affecting phosphorylation of LTAg [143].

Although it is assumed that sTAg of HPyV possesses the same properties as SV40 sTAg, MCPyV sTAg exerts different functions. It can fully transform Rat-1 and NIH3T3 mouse fibroblasts and also interacts with PP4. This interaction has an inhibitory effect on the NFκB signaling pathway and promotes microtubule destabilization, cell mobility, and filopodium formation (reviewed in [144]).

### 3.2. Putative Phosphorylation Sites of SV40 sTAg

Reports describing SV40 sTAg as a phosphoprotein are lacking. Since the first 82 amino acids are shared with LTAg and these are not phosphorylated in LTAg, it is unlikely that phosphorylation of the N-terminal region of sTAg occurs. In agreement with this assumption was the finding that a fragment comprising the 82 N-terminal aa of LTAg (which is identical to sTAg) did not show phosphorylation [145]. However, it cannot be ruled out that the configuration of full-length sTAg allows phosphorylation of this part of the protein, whereas in LTAg, the three-dimensional structure prevents phosphorylation of the first 82 residues because of the accessibility/inaccessibility of protein kinases. Bona fide phosphorylation sites in the unique sequence of sTAg are not known. Putative phosphorylation sites for SV40 sTAg include S10, S22, S57, T81, S87, Y96, S108, T167, and T168 (Appendix A). PKA (S10) and PKC (T57 and T167) may mediate the phosphorylation of these sites as their predicted score is relatively high (>0.750; with 0.000 = no similarity with the protein kinase consensus and 1.000 = completely identical to the consensus motif).

Phosphorylation of HPyV sTAg has, to the best of our knowledge, not been reported. Predicted phosphoacceptor sites with putative protein kinases that may mediate their phosphorylation are summarized in Appendix A. Conserved residues between all HPyVs or between species from the Alpha-, Beta-, and Deltapolyomavirus genera are depicted in Appendix A. Residue T156 in SV40 sTAg is conserved in all HPyV as either T or S, except in JCPyV sTAg, which has A in this position. In SV40, T156 is part of a PKC consensus motif, whereas in BKPyV, the corresponding residue (T154) is not part of a protein kinase recognition site. The corresponding site in KIPyV (T170), WUPyV (T174), and STLPyV (S167) is a putative target for PKC (Appendix A). Alpha- and Betapolyomaviruses have conserved S or T in sTAg: S86 and S92 in NJPyV and TSPyV, respectively; T90 in MCPyV and HPyV10, T91 (HPyV6, HPyV7, HPyV9, QPyV), and T92 in STLPyV. However, no or weak consensus motifs are found for protein kinases, except for T90 in MCPyV sTAg and T91 in HPyV6, which might be sites for PKC (Appendix A). SV40 sTAg Y139 is conserved in sTAg of all human Alpha- and Betapolyomaviruses but is not predicted to be phosphoacceptor site for the protein kinases ATM, CDK1, CDK5, CKI, CKII, CaMKII, DNAPK, EGFR, GSK3, INSR, p38^MAPK^, PKA, PKB, PKC, PKG, RSK, and SRC used in the Netphos 3.1 algorithm [72,73].

The residues Y96 and S108 are conserved in sTAg of BKPyV and JCPyV. T167 is conserved in all human Betapolyomaviruses (BKPyV, JCPyV, KIPyV, WUPyV), in the Deltapolyomaviruses HPyV6, HPyV10, QPyV, and HPyV7 (in the latter residue 167 is Ser), and in the Alphapolyomavirus MCPyV.

The residues T156L157 in SV40 sTAg are conserved in the corresponding sites of the other HPyVs as T/SL/F, except JCPyV, which contains the amino acid sequence AL (Appendix A).

## 4. Gaps in Knowledge and Future Research Directions

Our knowledge of HPyV LTAg and sTAg phosphorylation is, at best, limited. Phosphorylation of these PyV regulatory proteins has mainly been studied in SV40 and mouse polyomavirus (MPyV). Most of these studies were performed in the 1980s–1990s and mainly identified phosphoacceptor sites based on Edman degradation of ^32^P-labeled proteins, whereas protein kinases that catalyze phosphorylation are poorly characterized [74,75,76,146,147]. Comparing the known phosphorylation sites in SV40 and MPyV with the amino acid sequences of HPyV LTAg revealed that only Thr corresponding to T124 in SV40 LTAg (T278 in MPyV; [144]) is conserved (Appendix A). As phosphorylation of this residue is crucial for SV40 and MPyV DNA replication [86,147], it may suggest a similar role for the HPyVs. Neither phosphorylation of SV40 nor MPyV sTAg has been reported, nor has this posttranslational modification been examined in HPyV sTAg. Currently, mass spectrometry (MS) is the method of choice for the identification of phosphomodification in a protein [148]. Such studies have only been performed on MCPyV LTAg and in vitro PKA-phosphorylated peptides derived from MCPyV LTAg [132,133,135]. Therefore, mass spectrometry studies on the LTAg and sTAg from HPyVs should be performed. This will allow identification of phosphoacceptor sites. Phosphospecific antibodies can then be generated to confirm the phosphorylation sites, to study subcellular-dependent and cell-cycle-dependent phosphorylation, and to compare the phosphorylation state of these proteins in healthy and diseased tissue. The latter may provide information about whether aberrant phosphorylation of LTAg and/or sTAg plays a role in HPyV pathogenesis and whether the phosphorylation pattern of LTAg/sTAg can be used as a biomarker. Studies with mutant proteins in which phosphoacceptor sites are substituted may provide information of the biological importance of these phosphoresidues. Another almost untouched research field is the characterization of protein kinases and protein phosphatases that are responsible for phosphorylation and dephosphorylation. Protein kinase analysis using purified LTAg and sTAg and a panel of protein kinases and protein phosphatases may lead to the identification of protein kinases and protein phosphatases responsible for LTAg and sTAg (de)phosphorylation. Phosphorylation of HPyV LTAg and sTAg may be sequential, where a phosphorylation is required before a subsequent phosphorylation can occur. Because posttranslational modifications, including phosphorylation, of viral proteins can enhance the pathogenic properties of the virus [19,21,149], it is highly relevant to study these in HPyVs, which are involved in diseases such as nephropathy, progressive multifocal leukoencephalopathy, and malignancies [127,130,150]. Our understanding of the phosphorylation of the LTAg and sTAg of HPyVs may help to develop new therapeutic strategies, and the phosphorylation state of these proteins may serve as clinical tools for disease diagnosis, prognosis, and treatment response assessment.

## Figures and Tables

**Figure 1 viruses-15-02235-f001:**
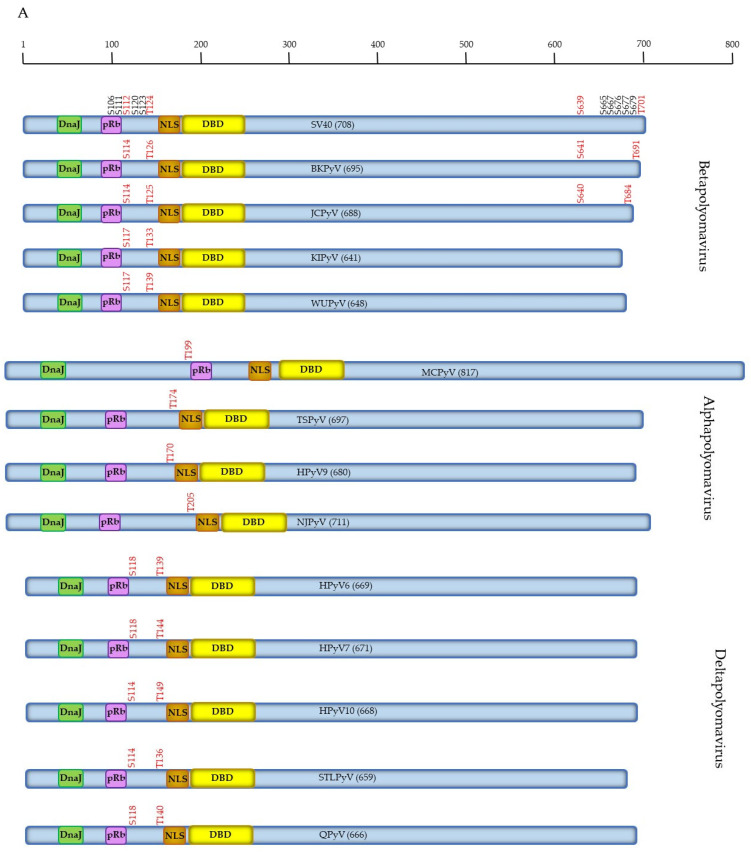
Conserved putative phosphoacceptor sites in LTAg and sTAg of HPyV. (**A**) The LTAg of SV40 and HPyV with the number of amino acid residues in parenthesis. The conserved Hsc70 binding domain (DnaJ), retinoblastoma family members binding motif LXCEX (pRb), the nuclear localization signal (NLS) and the DNA binding domain (DBD) are depicted. The proven phosphorylation sites in SV40 are shown on top of LTAg. The conserved residues and their corresponding residues in the LTAg of HPyV are shown in red. (**B**) Conserved putative phosphoacceptor site in SV40 and HPyV sTAg. SV40 T156 is conserved in all HPyVs except JCPyV. The number of amino acids is shown in parenthesis. The conserved Hsc70 binding domain (DnaJ) is indicated. The Betapolyomaviruses are shown first because they are most closely related to SV40.

**Table 1 viruses-15-02235-t001:** Functional phosphorylation sites in SV40 LTAg and effect of mutations in these sites compared to wild-type SV40 LTAg.

Residue Mutant	Protein Kinase	Support of Virion Production ^1^	Transforming Activity ^2^	DNA Rep ^3^	Site I Binding ^4^	Site II Binding ^4^	Expression Levels ^5^	Nuclear Localization ^6^	ATPase Activity ^7^	References
S106A	CKI, CKII, GSK3	comparable	comparable	same	same	same	slightly reduced	same	intact	[74,75,78,79,80]
S106F	CKI, CKII, GSK3	NT ^8^	reduced	same	NT	NT	NT	NT	NT	[74]
S111A/ S112C	CKII, DNAPK	reduced	comparable	reduced	same	same	same	reduced	intact	[75,78,79,81,82]
S112D	CKII, DNAPK	comparable	comparable	comparable	same	same	NT	enhanced	NT	[74,75,79,81,82]
S120A	ATM, CKI, DNAPK	none	comparable	none	same	same	same	reduced	intact	[74,77,78,80,82,83,84,85]
S123A	CKI, DNAPK	none	comparable	none	same	enhanced	same	Same	intact	[74,75,77,78,79,82,83]
T124A	CKI, CDK1	none	comparable	none	same	none	slightly reduced	increased	intact	[74,75,78,79,86]
T124E	CKI, CDK1	not tested	comparable	low	same	none	same	same	intact	[78]
S639A		enhanced reduced	enhanced	reduced	same	same	same	same	intact	[74,78,79]
S665	DNAPK	NT	NT	NT	NT	NT	NT	NT	NT	[82]
S667	DNAPK	NT	NT	NT	NT	NT	NT	NT	NT	[82]
S676A ^9^	CKI	enhanced reduced	enhanced	reduced	reduced	same	same	same	intact	[74,75,77,78,79]
S677A	CKI, DNAPK	reduced	reduced	same	none	same	same	same	intact	[74,75,78,79,82,83,87]
S679A	CKI	reduced	reduced	enhanced	same	enhanced	same	same	intact	[74,75,78,79,83,87]
T701A		comparable	comparable	same	same	same	same	same	intact	[74,78,79]

^1^ Virion production in TC7 monkey cells compared to wild-type LTAg; ^2^ Rat-2 cells compared to wild-type LTAg; ^3^ viral DNA transfected in monkey cell lines or purified LTAg added to cell-free systems prepared from, e.g., HEK293 or Raji cells; ^4^ in vitro binding of immunoprecipitated LTAg or baculovirus-expressed recombinant LTAg to plasmid DNA containing either site I or site II; ^5^ Western blot of different cell types transfected with LTAg expression vectors; ^6^ immunostaining with anti-LTAg antibody or GFP fusion proteins; ^7^ LTAg incubated with ^32^P-dATP, and free ^32^P measured; ^8^ NT: not tested; ^9^ excluded as phosphorylation site by [74].

## Data Availability

Not applicable.

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
