# Peer review of "Phosphorylation of Human Polyomavirus Large and Small T Antigens: An Ignored Research Field"

_viruses, 2023, doi:10.3390/v15112235_

Round 1
Reviewer 1 Report
Comments and Suggestions for Authors
This manuscript contains a comprehensive review of phosphorylation sites on the large tumor antigen and small tumor antigen from various human polyomaviruses. The function of phosphorylated resides are examined and listed in tables. The authors also contribute some putative phosphorylation sites on these proteins using Netphos 3.1. This review requires minor revisions before acceptance of the manuscript.
- The thresholds and input parameters that were used to predict putative phosphorylation sites using Netphos 3.1 should be described (either in the manuscript or supplemental) so that others can repeat and examine these phosphorylation sites using the program. The scores reported in Supp Table 1 should be also defined in the supplementary for ease of access and comprehensibility, not just the main text. However, it is still questionable if these tables are necessary since none of (or some of) these were experimentally proven. The score does not mean anything. Since it is a prediction it can really mislead unless it is examined.
- In Table 1, mutation S112D says "not tested" for the last three columns. "not tested" should be changed to NT so that the table can remain consistent.
- In Table 1, the nuclear localization of SV40 LT with numerous phospho-mutants is stated as the "same". However, from lines 132-162, there are numerous mutations that are stated that can increase or decrease the accumulation of LT in the nucleus. Therefore, there are some discrepancies between the text and the table.
- In line 363, should S293A be changed to S239A?
- For the section on MCPyV LT phosphorylation, it might be better to separate full-length LT from truncated LT into separate paragraphs/sections for more clarity between the two proteins.
Author Response
Reviewer 1
This manuscript contains a comprehensive review of phosphorylation sites on the large tumor antigen and small tumor antigen from various human polyomaviruses. The function of phosphorylated resides are examined and listed in tables. The authors also contribute some putative phosphorylation sites on these proteins using Netphos 3.1. This review requires minor revisions before acceptance of the manuscript.
We thank the reviewer for the useful comments and respond them point-by-point. By doing so, we think that it has improved the original manuscript.
The thresholds and input parameters that were used to predict putative phosphorylation sites using Netphos 3.1 should be described (either in the manuscript or supplemental) so that others can repeat and examine these phosphorylation sites using the program. The scores reported in Supp Table 1 should be also defined in the supplementary for ease of access and comprehensibility, not just the main text. However, it is still questionable if these tables are necessary since none of (or some of) these were experimentally proven. The score does not mean anything. Since it is a prediction, it can really mislead unless it is examined.
Response: We agree with the reviewer that the prediction program used to identify possible phosphorylation sites and protein kinases in the LTAg and sTAg of human polyomaviruses must absolutely be supplemented with experimental proof. We discuss this in section 4: “Gaps of knowledge and future research directions”. The Netphos 3.1 algorithm is a free on-line program that can be used by anybody. It only requires copying the amino acid sequence of the protein of interest. Therefore, and because the reviewer indicates that these tables are maybe not necessary, we feel that describing the thresholds and input parameters are superfluous.
We have added the definition of score in the Supplementary Tables 1 and 2 as requested by the reviewer.
- In Table 1, mutation S112D says "not tested" for the last three columns. "not tested" should be changed to NT so that the table can remain consistent.
Response: We have made this correction.
- In Table 1, the nuclear localization of SV40 LT with numerous phospho-mutants is stated as the "same". However, from lines 132-162, there are numerous mutations that are stated that can increase or decrease the accumulation of LT in the nucleus. Therefore, there are some discrepancies between the text and the table.
Response: Thank you for this important correction.
- Phosphorylation of S106 or of T124 both had a negative effect on nuclear import. However, when S106 was substituted by Ala (which does not allow phosphorylation), nuclear import of this mutants was as wild-type. Therefore, we wrote in the table “same”. However, the T124A mutant displayed increased nuclear import compared to wild-type LTAg. We have corrected this.
- Phosphorylation of S120 stimulated nuclear import and S120A had reduced nuclear import. We have corrected this.
- The double mutation of S111/S112 reduced nuclear import and we have also corrected this in the table.
- Mutation of S123 did not influence nuclear import.
- S112D LTAg mutant had increased nuclear import. We corrected this.
- In line 363, should S293A be changed to S239A?
Response: Thank you for pointing out this mistake. We have corrected S293A into S239A.
- For the section on MCPyV LT phosphorylation, it might be better to separate full-length LT from truncated LT into separate paragraphs/sections for more clarity between the two proteins.
Response: we agree with the reviewer and have divided the section of phosphorylation of Alphapolyomavirus LTAg (section 2.4.2 in the original manuscript) into following subsections:
2.4.2.1 Phosphorylation of Merkel cell polyomavirus full-length LTAg
2.4.2.2 Phosphorylation of Merkel cell polyomavirus truncated LTAg
2.4.2.3 Phosphorylation of Alphapolyomavirus LTAg

Reviewer 2 Report
Comments and Suggestions for Authors
The review by Moens and colleagues summarizes the current knowledge and functional implications of phosphorylation of the early gene expression products LT and sT of SV40 and relates this knowledge to the still very limited knowledge of phosphorylation of these proteins in human polyomaviruses. Thus, the review addresses a very interesting topic. The review mentions other early gene expression products of human polyomaviruses, such as 17KDa LT in SV40/BK/JC or ALTO and 57KDaLT in MCPyV, but remains short due to lack of knowledge. Overall, the review is sometimes difficult to read and often deviates slightly from the topic, especially in the introduction, which is less about the importance of PTM, e.g. phosphorylation, (one could imagine to describe examples), but goes into great detail about which PyV are counted as human polyomaviruses. This confuses the taxonomic classification with the debate whether SV40 is a human PyV, or at least it is written in a misleading way. This point, if there is to be a point here at all, should be a critical one. Also, the sentence in lines 97-99 is unclear to me as to what it adds at this position of the review.
The introduction could benefit from a description and discussion of past and current methods used to map phosphorylation sites. The functional significance of phosphorylation would also be a good section in the introduction.
The review has very detailed tables, which are very important, but the review would gain clarity by including a graphical representation of the conserved phosphorylation sites in LT and also in sT. Table 1 should also be improved in terms of describing the consequences of phosphorylation, perhaps adding a + or - only where there is a discrepancy with wt and otherwise annotating as is.
I wonder why mutants S665 and S667 are listed in Table 1 if these mutations have not been tested for the functions listed in the table? The categories describing the columns of the table need more clarification. Transforming activity in rat-1 cells; DNA replication (which assay? in vitro DNA replication or in cells?) expression levels (in which cells?); ATPase activity (which assay?). Site I and site II are mentioned in lines 87 and 89 without reference to the table.
Paragraph 2.3.3 is rather short; one can also imagine not to divide the review so strictly into SV40 and phosphorylation followed by much shorter chapters on alpha, beta, delta PyVs LT and phosphorylation. For SV40, the authors have done a nice job of dividing the chapter into functional chapters: DNA Replication, Protein-Protein Interaction, and Transformation. As a reader, it would be preferable to have a direct transfer of the content to the knowledge of human PyV at the end of each functional chapter.
The section on sT is very short compared to LT and is only described in detail for sT of SV40. Since the authors discuss the function of sT (of SV40) in the first paragraph, it would be important to note the different function of sT in SV40 and in MCPyV with respect to transformation.
The title could also be sharpened, since the review is mainly about current knowledge of phosphorylation of SV40 LT and sT and only comparatively discusses human PyVs early gene products and phosphorylation.
minor points:
line 85: please include motifs after 5'-GAGGC-3'
line 90: correct atttach to attach
line 120: please replace Proven phosphorylation sites with functional confirmed phosphorylation sites
columns of table1: please replace viability with support of virion production
line 121: subtitles of Table 1: Please include NT: not tested
Author Response
Reviewer 2
The review by Moens and colleagues summarizes the current knowledge and functional implications of phosphorylation of the early gene expression products LT and sT of SV40 and relates this knowledge to the still very limited knowledge of phosphorylation of these proteins in human polyomaviruses. Thus, the review addresses a very interesting topic. The review mentions other early gene expression products of human polyomaviruses, such as 17KDa LT in SV40/BK/JC or ALTO and 57KDaLT in MCPyV, but remains short due to lack of knowledge. Overall, the review is sometimes difficult to read and often deviates slightly from the topic, especially in the introduction, which is less about the importance of PTM, e.g. phosphorylation, (one could imagine to describe examples), but goes into great detail about which PyV are counted as human polyomaviruses. This confuses the taxonomic classification with the debate whether SV40 is a human PyV, or at least it is written in a misleading way. This point, if there is to be a point here at all, should be a critical one. Also, the sentence in lines 97-99 is unclear to me as to what it adds at this position of the review.
We thank the reviewer for the valuable and relevant comments and have responded point by point below.
The introduction could benefit from a description and discussion of past and current methods used to map phosphorylation sites. The functional significance of phosphorylation would also be a good section in the introduction.
Response: We have add a short description of the past and current methods used to determine phosphorylation sites. Following text was added:
Historically, phosphorylated residues were identified by using in vitro or in vivo 32P-labeled proteins. 32P-labeled peptides were separated by polyacrylamide gel electrophoresis or high-performance liquid chromatography and detected by autoradiography or scintillation counting. Edman degradation is then used to sequence the radiolabeled peptides to determine the phosphorylated residue(s). More recently, mass spectrometry is the method of choice for characterizing phosphorylated proteins. The protein of interest is enzymatically digested into peptides and analyzed using a mass spectrometer, in which the instrument records the mass-to-charge ratios of the various peptides. To identify a phosphorylated peptide, their mass-to-charge values of the peptides are compared to the expected mass-to-charge values of the peptides. Phosphorylated peptides have an increased mass of n x 79.9663 Da (i.e., mass of phosphate group; n=number of phosphates in the peptide). For reviews see [ref 15, ref 16].
The functional significance of phosphorylation is briefly discussed in the first paragraph of the introduction. The number of protein kinases, protein phosphatases, percentage of proteins in a cell that are phosphorylation and the roles phosphorylation plays in cellular processes are mentioned. To stress the importance of phosphorylation, we have added that aberrant expression or activity of protein kinases and protein phosphatases play a role in malignant and non-malignant diseases. We feel that a more detailed description is beyond the scope of this review.
The review has very detailed tables, which are very important, but the review would gain clarity by including a graphical representation of the conserved phosphorylation sites in LT and also in sT. Table 1 should also be improved in terms of describing the consequences of phosphorylation, perhaps adding a + or - only where there is a discrepancy with wt and otherwise annotating as is.
Response: We agree with the reviewer and have included a figure (Figure 1A in the revised manuscript) depicted SV40 and HPyV LTAg, the proven phosphoacceptor sites in SV40 LTAg and the conserved corresponding residues in HPyV LTAg. We also included a figure showing the conserved putative phosphoacceptor sites for sTAg (Figure 1B in the revised manuscript).
The table compares the function of the mutants with LTAg and differences are indicated. We feel that using “+” and “-“may be confusing because these symbols often refer to “present” or “absent”.
I wonder why mutants S665 and S667 are listed in Table 1 if these mutations have not been tested for the functions listed in the table? The categories describing the columns of the table need more clarification. Transforming activity in rat-1 cells; DNA replication (which assay? in vitro DNA replication or in cells?) expression levels (in which cells?); ATPase activity (which assay?). Site I and site II are mentioned in lines 87 and 89 without reference to the table.
Response: Indeed, the effects of mutations in residues S665 and S667 on the function of SV40 LTAg have not been studies, but these residues are mentioned because their phosphorylation has been confirmed and the protein kinase (i.e. DNAPK) that mediates their phosphorylation has been identified.
We have added more information in the footnotes of the Table on the most commonly used methods to investigate the effects of mutations on LTAg’s function.
Lines 89-92 in the original manuscript describe the origin of replication of SV40 and at this point the effect of mutations in LTAg phosphorylation sites on binding to these sites is not discussed. Therefore, we see no need to refer to Table 1.
Paragraph 2.3.3 is rather short; one can also imagine not to divide the review so strictly into SV40 and phosphorylation followed by much shorter chapters on alpha, beta, delta PyVs LT and phosphorylation. For SV40, the authors have done a nice job of dividing the chapter into functional chapters: DNA Replication, Protein-Protein Interaction, and Transformation. As a reader, it would be preferable to have a direct transfer of the content to the knowledge of human PyV at the end of each functional chapter.
Response: We agree with the reviewer, but while the functional implications of SV40 LTAg phosphorylation on nuclear translocation, replication, protein interaction, and transformation has been well studied, little or nothing is known on corresponding studies with the LTAg of HPyV. On the other hand, the effect of phosphorylation on transcriptional activity and apoptosis for MCPyV LTAg has been examined, by the equivalent studies for SV40 LTAg are lacking. So, it would be difficult to transfer this knowledge to the corresponding SV40 section in the manuscript. Reviewer 1 suggested us to divide the part on MCPyV LTAg into one subsection for full-length LTAg and one subsection for truncated LTAg, which we have done. So, if reviewer 2 accept, we would like to keep the outline of the manuscript as it is.
The section on sT is very short compared to LT and is only described in detail for sT of SV40. Since the authors discuss the function of sT (of SV40) in the first paragraph, it would be important to note the different function of sT in SV40 and in MCPyV with respect to transformation.
Response: we agree with the reviewer and have added following test:
Although it is assumed that sTAg of HPyV possess the same properties as SV40 sTAg, MCPyV sTAg exerts different functions. It can fully transform Rat-1 and NIH3T3 mouse fibroblasts and also interacts with PP4. This interaction has an inhibitory effect on the NFkB signaling pathway and promotes microtubule destabilization, cell mobility, and filopodium formation (reviewed in [148]).
The title could also be sharpened, since the review is mainly about current knowledge of phosphorylation of SV40 LT and sT and only comparatively discusses human PyVs early gene products and phosphorylation.
Response: We agree with the reviewer and have changed the title into:
“Phosphorylation of SV40 large and small t antigens and the human polyomaviruses orthologues: facts and predictions”
minor points:
line 85: please include motifs after 5'-GAGGC-3'
Response: We have made this correction.
line 90: correct atttach to attach
Response: Thank you for indicating this typo. We have corrected the mistake.
line 120: please replace Proven phosphorylation sites with functional confirmed phosphorylation sites
Response: Thank you. We have done so.
columns of table1: please replace viability with support of virion production
Response: We agree with the reviewer that “viability” is not the most accurate word to describe a virus. We have corrected as proposed by the reviewer.
line 121: subtitles of Table 1: Please include NT: not tested
Response: We have made this change.

Round 2
Reviewer 2 Report
Comments and Suggestions for Authors
The authors adressed all concerns raised previously, thus, significantly improving the review.